# Alternative Opportunities to Collect Semen and Sperm Cells for Ex Situ In Vitro Gene Conservation in Sheep

**Malam Abulbashar Mujitaba** [1,2], **István Egerszegi** [3,*], **Gabriella Kútvölgyi** [4], **Szabolcs Nagy** [5], **Nóra Vass** [2] **and Szilárd Bodó** [4]

1. Doctoral School of Animal Science, University of Debrecen, Böszörményi Street 138, H-4032 Debrecen, Hungary
2. Institute of Animal Science, Biotechnology and Nature Conservation, Faculty of Agricultural and Food Sciences and Environmental Management, University of Debrecen, Böszörményi Street 138, H-4032 Debrecen, Hungary
3. Department of Animal Husbandry and Animal Welfare, Institute of Animal Sciences, Szent István Campus, Hungarian University of Agriculture and Life Sciences, Páter Károly Street 1, H-2100 Gödöllő, Hungary
4. Department of Precision Livestock Farming and Animal Biotechnics, Institute of Animal Sciences, Kaposvár Campus, Hungarian University of Agriculture and Life Sciences, Guba Sándor Street 40, H-7400 Kaposvár, Hungary
5. Department of Precision Livestock Farming and Animal Biotechnics, Institute of Animal Sciences, Georgikon Campus, Hungarian University of Agriculture and Life Sciences, Deák F. Street 16, H-8360 Keszthely, Hungary
* Correspondence: egerszegi.istvan@uni-mate.hu; Tel.: +36-703618750

**Abstract:** The semen of domestic mammals is conventionally collected with an artificial vagina (AV) for artificial insemination (AI) or for short- or long-term storage. However, the procedure has certain drawbacks: animal training is not feasible in extensive animal care systems nor among wild species, as the trained animals sometimes fail to mount. Hence, there is a need for alternative semen collection methods. Electroejaculation (EEJ) and epididymal sperm recovery (ESR) are the two effective alternatives to AV. However, in recent years, animal welfare campaigners have called for the ban, in certain EU countries, of EEJ due to its inhumane nature. In this review, alternative methods of sperm collection (by EEJ and ESR, their qualities, and their freezing techniques) are highlighted, as well as the effects of EEJ on pre-freeze and post-thaw ram sperm quality parameters and the animal welfare progress made in EEJ between the 20th and 21st centuries. Additionally, the techniques for enhancing post-thaw sperm quality prior to freezing and for the freezing of EEJ and ESR spermatozoa are explored. ESR and EEJ are reliable alternatives to AV on certain occasions. EEJ is ideal for semen collection in wild or untrained animals, breeding soundness examinations, collection outside of the breeding season, and culling. At the same time, ESR is ideal in cases of castration, accidental death of elite sire, or postmortem for gene conservation purposes or assisted reproductive technologies (ARTs) studies.

**Keywords:** ram semen collection; semen quality; epididymis; animal welfare; artificial vagina; ex situ in vitro gene conservation

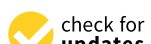



## 1. Introduction

Domestic animal production plays an integral role in the livelihoods of people in the developed and developing worlds by improving nutrition, ensuring food security, and promoting sustainability [1]. In developing countries, such as Nigeria, sheep are one of the major revenue contributors, so their economic relevance cannot be overstated. They serve as a ready source of income, meat, and milk for smallholder farmers due to their small body size, high reproductive performance, rapid growth rate, resilience to extreme weather conditions, and short generation interval compared to cattle [2,3].

The world sheep population reached a record of 1.266 billion head in 2022 [4]. However, autochthonous sheep genetic resources face a serious extinction threat across the globe due to a lack of proper identification and conservation. This has led to a wide paucity of information or records on the local sheep breeds. Records are the basis for breeding and gene conservation [5,6]. In 2018, altogether, 1164 local sheep breeds were registered in the Domestic Animals Diversity Information System. The risk status of all kinds of registered sheep breeds (1520) was very distinct, and more than two-thirds of them had an unknown risk status (780) or were classified as critical (97), endangered (218), and vulnerable (79) groups [7]. The 2022 Global Sustainable Development Goals reports on the number of animal genetic resources (AnGR) for food and agriculture secured in either the medium- or long-term conservation facilities on local sheep breeds and reveals a greater global and continental lack of information, rating them as being sufficient (SI), being not sufficient (NSI), or having no information (NI)/materials (NM). Globally, lack of material and information accounts for 87.22% (NI: 43.93% and NM: 43.29%): NSI, 8.39%; and SI, 4.39%. In North America and Europe, NM is 41.19%, NI is 38.45%, NSI is 12.61%, and SI is 7.75%, while in Sub-Saharan Africa, NM is 65.17%, NI is 32.58%, and SI is 2.25% [8]. Therefore, there is the need to strategize and invest more efforts to identify and conserve more local sheep breeds across the globe. The use of assisted reproductive technologies (ARTs), such as artificial insemination (AI), can help to achieve the targeted objectives. Therefore, this review is aimed at highlighting the historical background and presenting the status and possible alternative methods used to fulfil the conservation approaches applied to save these local sheep breeds all over the world.

The first step in the AI program is semen collection, which can be achieved in several ways: use of an artificial vagina (AV), including internal AV; transrectal massage; or electroejaculation (EEJ) [9]. In addition to the earlier mentioned methods, epididymal sperm recovery (ESR) also provides a good way of retrieving viable and promising quality spermatozoa for ART or gene conservation (GC) purposes [10,11]. In any case, most practitioners and ram and/or buck breeding stations prefer AV, but the animals need training [12]. The AV method also has certain welfare issues for both males (isolation stress during training) and females (restrained as dummies during collection) [13]. Moreover, in an extensive production system (e.g., nomadic and pastoral), or small-scale meat–milk producers' families, most farmers do not want to and/or cannot afford animal training nor AI [14]. Consequently, semen collection with AV is not successful, which has a negative effect on biodiversity conservation. EEJ and ESR are the two major alternatives to collecting semen for either ARTs or GC. These alternative semen collection methods can provide an interesting opportunity for conserving very important African sheep genotypes, such as the fat-tailed Damara sheep breed (a polyceraty gene carrying breed with multiple horns) from the southwestern region of Africa [15]; the endangered Namaqua Afrikaner [16]; and the largest sheep breed in Sub-Saharan Africa, Touabire [17].

*Historical Background*

The successful application of EEJ was first recorded in guinea pigs in 1922 by Batelli et al. [18]. In 1936, Gunn reported its application for the first time in ram semen collection [19], followed by other researchers: Bonadonna [20], Lambert and McKenzie [21], and Terill [22]. The stimulations of the previous version of the EEJ probe were 60 cycles of alternating current delivering up to 30 volts by the rectal probe and lumber electrode [23]. In 1945, Laplaud and Cassou invented a rectal probe with bipolar electrodes that did not need the external lumbar electrode for stimulations [24]. However, in 1948, Thibault et al. [25] modified the device by having 30 brass rings insulated from each other by ebony. It uses 60 cycles of alternating current, varying from 0 to 30, and then returns to 0 after every 3–5 s. A probe with a bipolar electrode was first used on rams in 1950 by Laurans and Clement [26]. The latest device in use is the three-longitudinal electrode probe that delivers a maximum of 10–15 volts of 30–50 sine or square waves [27]. EEJ can be used for non-trained and wild species such as the Spanish Ibex [28] and the white-tailed deer

(*Odocoileus virginianus*) [29]. It was first used in Hungary by Turányi János in 1970 at the Central Hungarian Artificial Insemination Station, Budapest, on imported British rams with accommodation problems [30]. EEJ was first used successfully on wild animals at the Budapest Zoo in 1977, i.e., on a wild goat (*Capra falconeri heptneri*) immobilized for hoof trimming [31], and, later, the process was used successfully on several mouflons [32]. Interestingly, with the EEJ technique, ram semen is collected with or without sedation (not allowed in certain EU countries) [33]. This depends on the operator and assistant's skill, their objectives, and the frequency of the semen collection. If needed, intramuscular injection of Xylazine or Acepromazine is effective [27]. The use of sedatives in carnivores was found to be associated with increased urine contamination in semen during EEJ. This has not been proven in ruminants [34]. However, Abril-Sánchez et al. [35] concluded that using sedatives or anesthesia in goat bucks decreased stress and pain responses and improved semen quality. Studies have revealed that semen collection methods have an impact on the sperm's quantitative and qualitative characteristics [11,36–39]. This calls for the need to explore more on the subject.

ESR is another way semen can be collected for ART or GC purposes. It bypasses all the welfare issues and obstacles associated with semen collection by AV or EEJ methods. It is a simple procedure, as viable and good quality spermatozoa can be retrieved postmortem or after emasculation of the male genital organs. It also permits the collection of spermatozoa from high genetic merit, endangered species, or zoo animals that die suddenly. Studies in rams revealed that it is possible to recover post-thaw viable epididymal spermatozoa 48 h postmortem [40,41] and 36 h post-orchiectomy in horses [42]. Moreover, it can be used successfully for either in vivo or in vitro fertilization [36]. Inseminations with epididymal sperm (EPS) resulted in good pregnancy (78–80%) and litter size (80–95%) in sheep [43], stallions [44], and cattle [45]. It therefore provides an excellent, simple, and economical means of collecting male gametes for ART and GC purposes. This review intends to explore the alternative methods of semen collection in sheep for ART and GC purposes and proffer strategies for compensating weak or poor-quality post-thaw sheep semen samples to achieve good fertility results. Furthermore, we consider pre-freeze and post-thaw parameters of spermatozoa collected with AV as a reference to enable us to compare with that of the spermatozoa collected by the alternative methods (EEJ and ESR) of semen collection.

## 2. Alternative Semen Collection Methods

### 2.1. Semen Collection by Electroejaculation

The EEJ procedure involves administering low-voltage, low-current electrical impulses to the male rectum, using a device called an electro-ejaculator (a transrectal probe equipped with an electrode), which leads to penile erection, emission, and subsequent ejaculation [33,46]. There are two types of ventral probes used in bulls: the segmented (allows operators to reduce stimulation of non-target tissues in the intrapelvic area) and non-segmented. The segmented probe has eight electrode segments: three short caudal ones for inducing penile erection (mostly used in younger bulls), three short middle ones for semen emission, and two short cranial segments (mostly activated in older bulls) which are used when EEJ is not elicited [9,47]. The most commonly used devices are Bailey and Ruakura ram probes. The Ruakura probe delivers 10–15 V output; when the rectum is dry, 15 V is recommended [27]. The probe should be lubricated before being inserted 15–20 cm [48]. Interestingly, semen collection by EEJ is independent of the ram's sexual desire [22]. It only takes 3–5 stimulations of accessory glands (10 to 15 s) for the ram to ejaculate. The stimulation is applied with the probe in a rhythmic on/off sequence (3–5 s on and 5–15 s off), with gentle downward pressure toward the pelvic floor [27]. The latest device is the Lane ram ejaculator (allows programmed or manually increasing electrical charge), which has a high/low switch (the high setting is more appropriate for rams, while the low one is more appropriate for goat bucks). Unlike the Bailey and Ruakura, with the Lane ram ejaculator, the prostate needs to be massaged by using the device 8–10 times. The stimulation is also applied in a rhythmic on/off sequence (4–6 s on and 5–8 s off, including

massage again during the resting period) (*Instructions of Lane Manufacturing Inc.*) [49]. Using a three-electrode probe (250 mm × 30 mm) connected to the electro-ejaculator, allowing voltage and amperage control, ejaculation is achieved at an average value of 4 V and 90 mA at about 3 min [50]. It is a straightforward procedure with the correct device [48]. The technique of EEJ facilitates and eases semen collection from many untrained rams or wild males for breeding or GC purposes. For detailed EEJ procedure in rams, see [50]. However, this technique is associated with certain drawbacks, part of which is that EEJ results in lower recovery (80.0% of the cases or 80.0% efficiency) than the AV method (100.0% efficiency) due to urine contamination or lack of response to the EEJ stimulation [37]. The major concerns or issues (distress and animal welfare, sperm quality, and reproductive performance) associated with semen collection by EEJ are explored in this review.

2.1.1. Distress and Animal Welfare Issues Regarding Electroejaculation

The AV technique has some drawbacks: certain trained rams can refuse to serve out-of-estrus or dummy ewes, which limit their use for breeding [22,23]. EEJ can be employed in such circumstances. It is used in humans (people with disabilities) and is widely practiced in small ruminants to collect semen, most notably for breeding soundness examinations. However, many studies have raised concerns over the effects of the procedure on animal welfare [51]. Between 1991 and 1995, the EU banned the importation of frozen semen collected by EEJ because of its inhumane nature [13,52,53]. Additionally, semen collection by EEJ is currently banned in several European countries [9,52,53], such as the Netherlands and Denmark. For regulations regarding semen collection, packaging, handling, transportation and distribution, and application in EU member states, see [51,54]. Animal welfare campaigners in Europe have been calling for the procedure to be banned [34]. In the UK, EEJ must be conducted under the supervision of a veterinary surgeon [51].

These techniques lead to a stress response when used frequently, with adverse effects on animal welfare [55]. The main indicators of distress associated with EE are vocalization (immediate response to pain); others include physiological indicators (blood cortisol, progesterone, and heart rate) [9,53]. According to Falk et al. [53] and Etson et al. [47], blood progesterone is a more sensitive indicator of stress due to EEJ in bulls. A study in rams revealed a significant relationship between vocalization and pain during EEJ [55]. Similarly, EEJ affects vital parameters (heart and respiratory rate increased), the concentrations of certain hormones (cortisol increased and testosterone decreased), and several blood parameters (glucose, total protein, and creatine kinase increased, while hematocrit, hemoglobin, red blood cell, and alkaline phosphatase decreased). However, it does not affect albumin and white blood cell concentrations. The changes in the vital parameters, cortisol, and blood parameter concentrations all returned to the basal levels at about 60-, 30-, and 120-min post-EEJ [55]. This suggests that the stress induced by the technique is temporary. It affirms the reports of Evans and Maxwell [48], which demonstrate that, apart from discomfort and muscular contractions, the procedure has no permanent ill effects. A study conducted by Stafford [34] revealed that part-shearing (one-eighth of the wool was removed each time) is more aversive than EEJ (transit time was significantly lower for EEJ than for part-shearing). However, when a ring-type electrode probe was used, the procedure was aversive. The duration and intensity of electrical stimulation and the number of animals affected can be used to assess animal welfare, while the probe type and stimulation pattern are used to determine the area of tissue affected. Ungerfeld et al. [56] investigated whether pleasurable stimuli, such as brushing, would aid in lessening the aversiveness of EEJ in rams. The authors concluded that there is no clear relationship. Anesthesia and analgesia can help reduce the pain and discomfort caused by the procedure [51]. Depending on the animal species, using anesthetics (increases risk of death), sedatives (lower risk of death, less stress, and fewer practical problems), or hormones (oxytocin, PGF2$\alpha$ analogues, or a combination of the two) can improve sperm quality and reduce stress and pain associated with EEJ [27,33,35]. When anesthesia is used, the ruminant animal needs to have fasted for 12–24 h prior to collection to avoid regurgitation and tympanism [57,58]. Similarly, vita-

mins can be used to reduce oxidative stress and improve sperm quality [59]. According to Abril-Sánchez [35], the sedated goat buck group was less stressed by EEJ, recovered faster, and produced a higher percentage of spermatozoa with the functional plasma membrane and normal morphology than the anesthetized group (due to high ketamine doses) and the control groups. Recently, there has been a call to use a less painful alternative to EEJ: the modified-EEJ technique in bulls [60] or the transrectal ultrasonic-guided massage of accessory glands in domestic and wild small ruminants (faster and requires fewer electrical pulses than EEJ, or none). However, it must be performed by experienced personnel [33]. More studies are needed on the two earlier-mentioned techniques in small ruminants.

Interestingly, from this review, it was observed that remarkable progress has been achieved in the EEJ procedure regarding animal welfare through improving the nature of the stimulation device (the probe), stimulation duration, and the voltage used (Table 1). The 21st-century device (three-longitudinal electrode probe) inflicts less pain and stress than the 20th-century device (probe type with ring electrodes) [22]. Similarly, Stafford [34] reported that stimulation using the probe with ring-type electrodes stimulates irrelevant nerves and skeletal muscles, inflicting severe pain and stress on the rams. Sperm concentration also improved by using the 21st-century probe. The highest concentration reported in the 20th century was $2.5 \times 10^9$ compared to $5.2 \times 10^9$ spermatozoa in the 21st century.

**Table 1. Electroejaculation** probe types and voltages used in different experiments.

| S/N | Century | Probe Description | Voltage (V) | Duration (s) | Rest Time (s) | Concentration ($10^6$/mL) | References |
|---|---|---|---|---|---|---|---|
| 1 | 20th | Bipolar electrode | 30 | 5 | 5 | 1400 | [61] |
| 2 | | Bipolar electrode | 30 | 5 | 5 | 229 | [22] |
| 3 | | Bipolar electrode | 30 | 8,5, and 3 | NA | 2500 | [61] |
| 4 | | Bipolar electrode | 8 | 40 c/s | NA | 1100 | [54] |
| 5 | | 2 Brass RE 22 cm L × 2.5 cm D | 5.1 | 3 | 7 | 1900 | [62] |
| 6 | | Ruakura probe | 11 | 10 | 0.02 | 2400 | [63] |
| 7 | 21st | 3-LE 22 cm L × 2.5 cm D | NA | 2 | 2 | 5200 | [37] |
| 8 | | 3-LE 23 cm L × 2.5 cm D | 8 | 3 | NA | NA | [55] |
| 9 | | 3-LE 25 cm L × 3 cm D | 4 | NA | NA | NA | [50] |
| 10 | | RE 16.5 cm L × 1.7 cm D | 9 | 2–5 | 5 | NA | [64] |
| 11 | | 3-LE 22 cm L × 2.5 cm D | NA | 2 | 2 | NA | [65] |
| 12 | | 3-LE 22 cm L × 2.5 cm D | NA | 5 | 5 | NA | [11] |
| 13 | | 3-LE 35 cm L × 3.2 cm D | 5 | 5 | 5 | 4900 | [38] |
| 14 | | 3-LE 25 cm L × 2.5 cm D | 9 | 5 | 5 | NA | [66] |

D, diameter; L, length; LE, longitudinal electrode; RE, ring electrode; and NA, no available data.

### 2.1.2. The Quality of Electroejaculated Ram Spermatozoa

Several studies revealed the impact of collection methods on semen pre-freeze and post-thaw characteristics. In both rams and goat bucks, the ejaculate collected by EEJ has a higher volume and lower concentration than the AV-collected [38]. Matthews et al. [67] reported no difference in ram sperm morphology due to collection methods. Quinn et al. [68] concluded that EEJ seminal plasma increased ram spermatozoa's susceptibility to cold shock, with the AV-collected spermatozoa being more resistant. Similarly, Álvarez et al. [69] reported a more inferior post-thaw sperm quality collected by EEJ due to its higher sensitivity to high glycerol concentration. In contrast, the thawed and incubated EEJ spermatozoa presented a significantly higher percentage of intact acrosomes, mitochondrial membrane potentials, and cleavage rate following heterologous IVF than the post-mortem collected spermatozoa [70]. These findings are in accordance with the report of Marco-Jiménez et al. [37], who showed that the EEJ-collected ejaculates presented significantly higher post-thaw intact acrosomes and live cells non-capacitated than the AV post-thaw ejaculates. According to Marco-Jiménez et al. [71], using EEJ in semen collection might change the secretory functions of one or more of the accessory sex glands. It modifies the seminal plasma composition. It is in line with the findings of Barrios et al. [36] and Maxwell et al. [72] that the collection method affects plasma volume and compositions such as proteins. Seminal plasma composition changes induce alterations in the sper-

matozoa structure, making its membrane less resilient [71,72]; this negatively affects the freezability, viability, and fertility of the sample [69]. Moreover, one of the drawbacks of the EEJ technique is the high tendency of urine and/or accessory gland secretion contamination. However, changing the collecting tube between emissions could minimize the risk of urine contaminations [70]. In contrast, semen collection using EEJ increases the concentration of low-molecular-weight proteins (RSVP14 and RSVP22 [65], as well as RSVP14 and RSVP20 [36]) in the seminal plasma, and this may positively affect the freeze–thawing resistance of the spermatozoa. However, the seminal plasma proteins are also affected by season, with autumn and winter being the most effective in increasing the motility of frozen–thawed ram spermatozoa [73]. Therefore, post-thaw semen quality differs depending on the collection method [11,74] and season [73].

The EEJ-collected spermatozoa were of higher quality and more resistant to cryopreservation than the AV-collected spermatozoa [11,37]. Table 2a,b show the collection method's impact on certain important pre-freeze and post-thaw ram sperm quality parameters, respectively. The studies reviewed herein reveal that the fresh AV-collected ejaculates were of higher concentration than those collected by EEJ. Conversely, the EEJ-collected ejaculates present a higher volume of spermatozoa, and almost all the other quality parameters show no significant difference compared to the AV-collected spermatozoa. In the post-thaw samples, most of the parameters show no significant difference between the two collection methods. This suggests that the EEJ technique can be used as a substitute for the AV techniques. Although not significantly different, the AV frozen–thawed ejaculates present a higher percentage of total motility, progressive motility, and linearity. Linearity was reported to be the best and only reliable predictor of fertility (cleavage rate) [70].

**Table 2.** (**a**) Effects of collection methods on fresh ram semen's quality parameters. (**b**) Effects of collection methods on post-thawed ram semen's quality parameters.

| (a) | | | | | | | | |
|---|---|---|---|---|---|---|---|---|
| Fresh | | | | | | | | |
| Collection Methods | Volume | Concentration (/mL) | Functional Mitochondria | Intact Plasma Membrane | Abnormal Sperm (%) | Total Motility (%) | Progressive Motility (%) | References |
| EEJ | 1.46 | $1.4 \times 10^9$ [a] | NA | NA | NA | NA | NA | [61] |
| AV | 1.18 | $1.9 \times 10^9$ [b] | NA | NA | NA | NA | NA | |
| EEJ | 1.12 | $1.1 \times 10^9$ [a] | NA | NA | 15.3 | 76 [a] | NA | [22] |
| AV | 0.71 | $2.7 \times 10^9$ [b] | NA | NA | 16.4 | 84 [b] | NA | |
| EEJ | 1.4 [a] | $2.5 \times 10^9$ [a] | NA | NA | NA | NA | NA | [75] |
| AV | 1.0 [b] | $4.3 \times 10^9$ [b] | NA | NA | NA | NA | NA | |
| EEJ | 1.3 | $1.1 \times 10^6$ [a] | NA | NA | 4.4 | NA | NA | [67] |
| AV | 1.1 | $1.7 \times 10^6$ [b] | NA | NA | 5.7 | NA | NA | |
| EEJ | 1.1 | $2.34 \times 10^9$ [a] | NA | NA | 4.0 | 77.18 [a] | NA | [76] |
| AV | 1.2 | $3.10 \times 10^9$ [b] | NA | NA | 3.9 | 86.7 [b] | NA | |
| EEJ | 1.1 | $5.2 \times 10^9$ [a] | NA | NA | 8.1 | 71.9 | NA | [37] |
| AV | 1.2 | $6.2 \times 10^9$ [b] | NA | NA | 9.1 | 69.8 | NA | |
| EEJ | NA | NA | NA | NA | NA | 74.3 | 46.6 [b] | [50] |
| AV | NA | NA | NA | NA | NA | 77.5 | 59.0 [a] | |
| EEJ | 3.99 [a] | NA | 47.7 [a] | 71.3 [a] | NA | 81.7 | 77.5 | [65] |
| AV | 1.74 [b] | NA | 16.7 [b] | 49.5 [b] | NA | 71.7 | 78.3 | |
| EEJ | NA | NA | 33.2 | 68.0 [a] | NA | 67.5 [a] | 76.0 [a] | [11] |
| AV | NA | NA | 27.4 | 47.7 [b] | NA | 46.7 [b] | 52.0 [b] | |
| EEJ | 1.25 [a] | $1.7 \times 10^9$ [a] | NA | 55.3 [a] | NA | 76.04 | 67.52 [a] | [77] |
| AV | 0.79 [b] | $4.9 \times 10^9$ [b] | NA | 70.5 [b] | NA | 74.79 | 53.67 [b] | |

**Table 2.** *Cont.*

| | | | | | | | | | | | |
|---|---|---|---|---|---|---|---|---|---|---|---|
| **(b)** | | | | | | | | | | | |
| | | | | | Post-Thaw | | | | | | |
| Collection Methods | Viability (%) | TM (%) | PM (%) | LN (%) | LR (%) | Linearity (%) | IPM (%) | IA (%) | DA (%) | FM (%) | References |
| EEJ | 30.4 | 39.6 | NA | 24.2 [a] | 1.4 [a] | 60.3 | NA | 26.0 [a] | 75.4 | NA | [37] |
| AV | 28.5 | 40.3 | NA | 20.7 [b] | 3.4 [b] | 54.1 | NA | 21.2 [b] | 72.6 | NA | |
| EEJ | 24.95 | 28.5 | 7.67 | NA | NA | 39.37 | NA | 21.65 | 31.1 [a] | NA | [77] |
| AV | 25.39 | 29.92 | 8.21 | NA | NA | 40.80 | NA | 23.31 | 43.5 [b] | NA | |
| EEJ | 25.5 | 48.7 [b] | 27.6 [b] | NA | NA | 65.3 | NA | NA | 34.2 | NA | [69] |
| AV | 29.3 | 50.1 [a] | 38.7 [a] | NA | NA | 68.4 | NA | NA | 33.5 | NA | |
| EEJ | NA | 34.8 | 36.2 | NA | NA | NA | 43.3 [a] | NA | NA | 20.6 [a] | [11] |
| AV | NA | 36.4 | 35.6 | NA | NA | NA | 41.1 [b] | NA | NA | 29.1 [b] | |

AV, artificial vagina; DA, dead/damaged acrosome; EEJ, electroejaculation; FM, functional mitochondria; IA, intact acrosome; IPM, intact plasma membrane; LN, live cells non-capacitated; LR, live acrosome reacted; NA, no data available; PM, progressive motility; TM, total motility. Means in the same column between two collection methods with different superscripts differ significantly ($p < 0.05$).

Most of the studies reviewed herein did not consider the effects of collection methods on certain quality parameters, such as the mitochondria and membrane integrity of the freshly collected ejaculates. On the other hand, acrosome reaction and capacitation status were the parameters not evaluated in the post-thaw samples by the majority of the studies. Therefore, future studies should consider assessing more quality parameters in the fresh and post-thaw ejaculates. More studies are needed to ascertain these findings because there are limited studies on the effects of collection methods (EEJ vs. AV) on the post-thaw quality or freezability of ram ejaculates.

### 2.2. Sperm Collection after Castration or Postmortem

The epididymis is an organ where remodeling of the sperm plasma membrane takes place, which gives it the motility and fertilizing ability [78]. Specifically, ram spermatozoa acquire motility and egg-binding ability in the *corpus* [79]. Sperm cells can be retrieved postmortem or after emasculating male genital organs from *cauda epididymis*. The procedure involves collecting the testes and epididymides (>100 g) [80], using a solution of 0.9% NaCl, 100 UI/mL penicillin G, and 100 mg/mL streptomycin sulphate. Testes can then be maintained at a temperature between 4 and 8 °C or room temperature (22 °C) and transported to the laboratory within 2 h postmortem, without any adverse effect on the sperm quality [50]. Upon removing the *tunica vaginalis*, the epididymis tube can then be separated from the testicle and washed with Dulbecco's Phosphate Buffered Saline (DPBS) at 37 °C. The *cauda* is then minced or sliced with a scalpel to retrieve the sperm cells [40,81]. Alternatively, it can also be retrieved by an incision method. Following isolation of the *cauda epididymis* from the tube, 6–8 longitudinal incisions can be made at the ventral part of the *cauda epididymis*. The incised *cauda epididymis* is then dipped into a 3 mL Tris buffer in a 35 mm Petri dish for 30 min for the spermatozoa to swim out into the buffer at room temperature [50,82] or 37 °C [41]. Finally, the incised epididymis can be rinsed with 1 mL buffer and washed twice in 4 mL buffer solution by centrifuging at $885\times g$ for 10 min. The pellet can then be diluted [82]. Instead of buffers, other authors used semen extenders maintained at 35 °C for 5 min [83,84]. In rabbits and larger mammals, spermatozoa can be retrieved by retrograde flow/flushing (less contamination) of the epididymis via the *ductus deferens* [85,86]. In smaller animals with tiny epididymides, floatation is an ideal technique [86]. EPS typically has a high percentage of distal cytoplasmic droplets; centrifuging or incubating it in a water bath (29–35 °C) for 30–60 min, using a glycerol-free extender, facilitates the spontaneous release of the droplets, and this helps change the sperm's circular movement pattern to rectilinear [86,87].

The Quality of Ram Epididymal Spermatozoa

The position of the testicle (right or left) was reported to have no significant effects on the quantity of ram EPS [80,88], but this is not the case in bulls [89]. Hence, the need for further research to investigate whether quality differences between the two cauda epididymal spermatozoa of the same ram exist, or if such differences are species-specific. However, animal age and both testicular and epididymal weight determine the total epididymal reserve and the content of the three different epididymal parts (*caput, corpus,* and *cauda*) [88]. In rams, if the epididymis cannot be processed immediately postmortem or after emasculation, it can be preserved at 5 °C for up to 24 h [40] or 48 h at 4 °C [41,83] and up to 7 days in mice [90], and viable spermatozoa can still be retrieved. However, increasing the storage time beyond the first 24 h postmortem can have an adverse effect on quality parameters [40]. EPS and its constituents differ slightly from ejaculated semen, lacking seminal plasma. Therefore, their use for AI or in vitro fertilization (IVF) requires specific procedures other than those used for ejaculated spermatozoa [91]. EPS usually has a smaller volume but is highly concentrated with a higher percentage of cytoplasmic droplets than the ejaculated semen [92]. However, the ejaculated ram semen contains higher reducing sugar (fructose) than the epididymal, which contains ascorbic acid as the primary reducing sugar. This enables the ejaculated spermatozoa to maintain better motility in anaerobic conditions [92]. Moreover, epididymal spermatozoa chromatin is less condensed than that of the ejaculated sperm, rendering it relatively more susceptible to external stimuli [93]. In contrast, ram EPS retrieved at 0 and 48 h postmortem presents similar fertilizing ability to ejaculated spermatozoa [40]. Similarly, a comparative IVF study in sheep with AV and EPS affirmed this finding [69,83]. Therefore, EPS can yield an outstanding result as the ejaculate. However, its quality is affected by the part or segment (*caput, corpus,* and *cauda*) of the epididymis from which the spermatozoa are retrieved. Ewes inseminated laparoscopically with EPS collected from distal and proximal *cauda epididymis* yielded a higher pregnancy rate, 80.0% (16/20) and 78.0% (29/37), respectively, than the ejaculated spermatozoa, 72.0% (21/29). At the same time, distal *caput* spermatozoa resulted in a 0.0% (0/8) pregnancy rate. Nevertheless, the spermatozoa from proximal *cauda* and ejaculate resulted in a significantly larger litter size, $1.87 \pm 0.70$ and $1.82 \pm 0.75$, respectively, than the distal *cauda* spermatozoa, $1.57 \pm 0.51$ [43]. Similarly, spermatozoa of higher quality were retrieved from *cauda* epididymides in horses [81]. This difference is due to the fact that as spermatozoa pass from the *caput* down to the *cauda* part of the epididymis, they undergo certain biochemical changes. Upon reaching the *cauda* segment, they encounter numerous substances, including immobiline, which significantly reduces their metabolism, resulting in minimized energy demand and less residue production. At this stage, their epithelium is not well enough developed to absorb waste, which is higher in the *caput* and slightly less in the *corpus* [81,94,95]. This implies that EPS quality is segment dependent. Therefore, the distal and proximal parts of the *cauda* epididymis should be the main targets for EPS retrieval.

## 3. Techniques for Enhancing Post-Thaw Sperm Quality Prior to Freezing

### 3.1. Treating Sperm Cells with Certain Biological Compounds or Chemicals

Sperm treatment with certain biological compounds or chemicals has been reported to improve its kinetic parameters and fertilizing ability. Bacterial lipopolysaccharide at a concentration below its spermicidal effect can change the progressive motility and other motion parameters of EPS that might result in infertility [96]. Similarly, Goodarzi et al. [97] concluded that treating ram EPS with low concentrations (1 mM) of exogenous cAMP but not cGMP improves the sperm motion parameters. In a similar study, a low concentration of L-arginine was found to have little effect on sperm motion parameters, while a high concentration has a suppressing effect [98]. Heparin treatment on ram spermatozoa did not significantly affect acrosome reaction rate. In contrast, calcium ionophore (A23187) increased the acrosome reaction rate of *cauda* epididymal and ejaculated spermatozoa but did not influence spermatozoa retrieved from the *corpus* and *caput* [79]. Supplementing

the semen extender with 5 mM of L-Glutamine (G8540) can significantly improve the motility and plasma membrane integrity of ram ejaculated spermatozoa [99]. Similarly, 5 or 10 µg/mL of quercetin in a semen extender can significantly increase the number of zygotes and morula- and blastocyst-stage embryos [100].

### 3.2. Treating Sperm Cells with Certain Ions

Treating the sperm cells with certain ions, such as potassium and phosphate, increases the fructose oxidation and aerobic fructolysis of ejaculated spermatozoa (increased metabolism and motility of spermatozoa). However, phosphate leads to higher fructolysis (10% more) than potassium and accumulated lactate, which appears more in washed than unwashed sperm cells. Phosphate has a similar effect on the epididymal spermatozoa, while potassium has no significant effect on fructolysis and fructose oxidation. In both ejaculate and epididymal spermatozoa, calcium ion treatment decreases sperm oxygen uptake and fructose oxidation. This effect is more evident when combined with phosphate ions [101]. Exposing ram EPS to seminal plasma has a marked stimulatory effect on motility with adverse effects on viability even when exposed for periods as short as 15 min [102].

## 4. Freezing of Electroejaculated and Epididymal Ram Spermatozoa

### 4.1. Freezing Electroejaculated Spermatozoa

The freezing of EEJ semen samples is fundamentally the same as the AV-collected semen. Generally, following concentration adjustment, ram semen is diluted to at least 1:8 with the selected extender, with ultimate dilution dependent on the manner of insemination (laparoscopic, cervical, or transcervical AI). Semen to be frozen in pellets (using a block of dried ice or solid $CO_2$ with a temperature of $-79$ °C) is generally diluted to just 1:3 to 1:4, depending on sperm assessment [27,48]. Dilution of fresh semen mainly employed either ultra-heat-treated milk; Dulbecco's Phosphate Buffered Saline supplemented with or without 2.0% equine serum or 10.0% FCS; or boiled milk (92–95 °C for 8–10 min cooled to 30 °C before use [27]. The major cryoprotectants in ram semen freezing are egg yolk (non-permeating stabilizer and protector of sperm membrane) and glycerol (permeating cryoprotectant). Other permeating cryoprotectants, such as ethylene glycol, acetamide, or dimethylformamide, can also be used [103,104]. For egg-yolk-based extenders, a 20.0% egg yolk solution was more suitable for freezing sperm regardless of its source. Glycerol at 4.0% improves post-thaw sperm quality collected by AV and EEJ. In recent years, there has been a call by researchers against the use of egg-yolk-based extenders. This was due to the wide variability of its components and microbial contamination risk, which might lead to endotoxin production, reducing the post-thaw viability and acrosomal integrity of sperm [105]. An alternative to the egg yolk is the plant-based cold-shock protector, lecithin. Commercially prepared semen extenders of different sources and/or compositions can also be used. They include soy-lecithin-based (Andromed®, Bioxcell®, Biociphos Plus®, and Botu-Bov®-soy lecithin), egg-yolk-based (Biladyl®, Botu-Bov® Triladyl®, and BullXcell®), milk-based (INRA96®), and protein-free (OptiXcell®) ones [106–110]. The final semen concentration depends on the insemination method. However, it is mainly maintained at $200 \times 10^6$ total spermatozoa per mL [50,111–113]. Semen freezing in straws can be achieved (i) manually in a Styrofoam box over liquid nitrogen vapor (static $LN_2$ vapor freezing) or (ii) using an automated programmable freezing machine [48,86,111]. Using a programmable freezer involves maintaining the following temperature profile: +4 °C, $-10$ °C (120 s), $-80$ °C (450 s), $-120$ °C (100 s), and $-140$ °C (180 s). After attaining the final freezing temperature at the stipulated time ($-140$ °C at 180 s), the semen straws are then plunged into LN2 ($-196$ °C) in a cryogenic container for permanent storage [114]. Nonetheless, both methods lead to good post-thaw motility and viability in rams. Conventionally, the semen is filled into the straws (usually 0.25 mL), and the filled semen is equilibrated in a refrigerator at 4–5 °C for 2 h. The freezing is achieved by placing a rack containing straws at 4 cm above the $LN_2$ surface for 8–10 min, using a Styrofoam box [48]. However,

for better cryopreservation results, the EEJ ejaculate needs an extender of lower glycerol concentration and osmolality compared to that used in freezing ESR spermatozoa [114].

*4.2. Freezing Epididymal Spermatozoa*

The freezing procedure for EPS is similar to that of AV- and EEJ-collected spermatozoa, but with slight differences: before dilution, EPS are usually washed with an isotonic buffer solution or m-PBS by single [41,101] or double [82] centrifugation at $500$–$600 \times g$ for 5–7 min or $885 \times g$ for 10 min, respectively. Regarding the permeating cryoprotectants, e.g., glycerol, instead of the 4.0%, which is ideal for AV- and EEJ-collected spermatozoa, the optimal level for the epididymis recovered samples is 8.0% [50,84]. This is in line with the findings of Martinez-Pastor et al. (2006), who showed that the ERS samples required an extender with higher glycerol concentration and osmolality than the EEJ-collected samples. The equilibration period is maintained at 4 or 5 °C for 2 h [41,70,83,84,100]. Recently, different authors have reported modifications in freezing ram epididymal spermatozoa by using 0.25 mL straws. However, although the equilibration temperature of 4 °C is maintained by all the authors, most of them modified one or two of the major stages of the conventional manual ram semen freezing technique by performing one or two of the following:

i. Maintaining the conventional equilibration duration of 2 h and extending the freezing duration at 4 cm above $LN_2$ to 20 or 30 min [40,82].
ii. Maintaining the conventional equilibration duration and freezing the straws at 3 cm above $LN_2$ for 10 or 15 min [100,115].
iii. Shortening the equilibration duration to 1 h and extending the freezing duration at 4 cm above $LN_2$ to 20 or 30 min [83,87].
iv. Extending the equilibration duration to 3 or 4 h [82,115].

In bulls, maintaining samples with a high percentage of spermatozoa having tail cytoplasmic droplets in a water bath, using a glycerol-free extender, for 30–60 min helps in the release of the droplets [87]. Moreover, using fructose as an energy source maintains a higher quality of ram *cauda* EPS [84,95].

In sheep, the EPS was reported to be more resistant to cold shock [83] and high temperatures (50 °C) than the ejaculated spermatozoa. In wildebeest antelope (*Connochaetes taurinus*), the cryo-survival of the epididymal sperm cells was found to be similar to that of the EEJ-collected [116]. However, in horses, the epididymal samples have a similar sperm concentration as the AV ejaculate; however, subjecting them to multiple freezing and thawing steps reduces their quality [117]. In contrast, an excellent post-thaw recovery rate of 87.1% total motility, 81.8% progressive motility, 74.4% normal acrosome, and 68.4% plasma membrane integrity from ram EPS were achieved [40]. Similarly, in a study conducted by Ehling et al. [87], fresh ram EPS motility was 79.7%, with 93.7% acrosome integrity. Their corresponding thaw values were 60.5% and 87.5%, respectively. Upon intrauterine AI, the lambing rate was found to be 87.5%. However, the important factors worth noting here are the slight modifications to the conventional freezing and thawing procedures:

i. Straws (fine paillette, 0.25 mL, Tiefenbach, Germany) were cooled from room temperature (20 °C) to 4 °C, at a rate of $-0.26$ °C/min.
ii. The equilibration period was shortened to 1 h instead of 2 h.
iii. Freezing of samples at 4 cm above the liquid nitrogen ($LN_2$) vapor was extended to 30 min as opposed to the normal period of 8–10 min.
iv. The thawing period was shortened to 17 s at 37 °C instead of 30 s at 37 °C.

Similarly, different studies revealed that EPS resulted in good-to-excellent pregnancy rates following AI: 92.0% in boar [118]; 58.8% in cattle [45]; 87.5%, 58.5%, and 55.0% in sheep [87,119,120]; 61.2% in goat [121]; 27.8% in stallion [44], and 75.0% in red deer [122].

In addition, frozen–thawed EPS collected from the epididymis stored at 5 °C for 24 h presents a similar fertilizing ability to that of frozen–thawed ejaculate [40]. Moreover, a study on European Mouflon (*Ovis musimon*) revealed that EPS have significantly higher overall post-thaw quality parameters (progressive motility, curvilinear velocity, straight-

line velocity, average path velocity, amplitude of lateral head displacement, linearity, and wobble) than ejaculated spermatozoa [116]. The AV-collected spermatozoa parameters were used in this review as a reference to enable a comparison of the cryopreservation response of AV and EPS. Assessing the recovery rates of the two crucial sperm motility and morphology parameters revealed that ESR spermatozoa present a better recovery rate for total and progressive motility, while AV-collected spermatozoa have slightly better intact acrosome and plasma membrane (Table 3). This contrasts with the findings of Goovaerts et al. [89], who compared the AV and EPS of the same bull and revealed that the AV spermatozoa has higher total motility and progressive motility, and that it moves more straight (higher straight-line velocity) with lower curvilinear velocity and amplitude than the EPS. Similarly, comparing the AV, EEJ, and EPS recovery rates reveals that EPS has better TM and PM (Table 3). Therefore, EPS is the best, most affordable, and most convenient source of male gametes for ARTs studies or GC purposes.

**Table 3.** Comparison of cryopreservation responses of artificial vagina, electroejaculation, and epididymal sperm recovered ram spermatozoa.

| | Pre-Freeze (%) | | | Post-Thaw (%) | | | Recovery Rate (%) | | | |
|---|---|---|---|---|---|---|---|---|---|---|
| **Parameters** | **AV** | **EEJ** | **ESR** | **AV** | **EEJ** | **ESR** | **AV** | **EEJ** | **ESR** | **References** |
| TM | 74.2 | 71.9 | 80.3 | 52.5 | 40.3 | 69.8 | 70.8 | 56,1 | 87.0 | [37,40,123] |
| | 77.5 | 76.0 | 79.9 | 59.1 | 28.5 | 62.0 | 76.3 | 37.5 | 77.6 | [50,77] |
| | 82.5 | 74.3 | 82.8 | 47.5 | 48.7 | 67.2 | 57.6 | 65.5 | 81.2 | [50,111,115] |
| | 87.6 | 67.5 | 77.5 | 63.2 | 34.8 | 53.3 | 72.1 | 51.6 | 68.8 | [11,82,124] |
| PM | | | | | | | | | | |
| | 70.8 | 67.52 | 68.4 | 45.4 | 7.7 | 55.8 | 64.1 | 11.4 | 82.0 | [40,77,124] |
| | 78.0 | 46.6 | 74.0 | 54.0 | 27.6 | 48.0 | 69.2 | 59.2 | 64.9 | [71,84] |
| | 59.0 | 76.0 | 57.6 | 38.7 | 36.2 | 41.5 | 65.6 | 47.6 | 72.0 | [11,50] |
| | 52.0 | NA | 66.4 | 35.6 | NA | 54.3 | 68.5 | NA | 81.8 | [11,50,115] |
| | NA | NA | 75.8 | NA | NA | 45.0 | NA | NA | 59.4 | [82,115] |
| IA | | | | | | | | | | |
| | 90.4 | NA | 84.3 | 73.1 | NA | 62.7 | 80.9 | NA | 74.4 | [40,111] |
| | 91.6 | NA | 91.8 | 80.6 | NA | 79.4 | 88.0 | NA | 86.5 | [82,123] |
| | NA | NA | 92.4 | NA | NA | 74.6 | NA | NA | 80.7 | [111] |
| IPM | | | | | | | | | | |
| | 84.6 | 55.3 | 84.2 | 60.9 | NA | 57.5 | 71.9 | NA | 68.3 | [40,77,111] |
| | 84.1 | 68.0 | 88.1 | 65.5 | 43.3 | 75.4 | 77.9 | 63.7 | 85.6 | [11,84] |
| | 66.9 | NA | 85.3 | 51.7 | NA | 61.1 | 77.3 | NA | 71.6 | [82,122,125] |
| | 47.7 | NA | 88.7 | 41.1 | NA | 53.6 | 86.2 | NA | 60.4 | [11,115,124] |
| | NA | NA | 86.4 | NA | NA | 57.2 | NA | NA | 66.2 | [125] |

AV, artificial vagina; EEJ, electroejaculation; ESR, epididymal sperm recovery; NA, no available data; TM, total motility; PM, progressive motility; IA, intact acrosome; IPM, intact plasma membrane.

## 5. Conclusions

Based on this review, EEJ and ESR are useful alternatives to the conventional semen collection method (AV) for ARTs studies or GC purposes. However, among the two alternatives (EEJ and ESR), ESR appears to be the most affordable and better alternative (higher recovery rates). Moreso, ESR permits the retrieval of male gametes in cases of culling, castration, slaughter, or accidental death of high genetic merit sire. Interestingly, the collected testicles can be stored in refrigerated conditions (4 °C) for up to 24 h before preservation, without adversely affecting the spermatozoa quality. The retrieved spermatozoa can be used fresh for direct AI or cryopreserved for later use. Therefore, it is an ideal source of male gametes for genome resource banking or ARTs studies.

There is no doubt that EEJ is stressful for the animals, but it is also an unavoidable procedure on certain occasions (semen collection in wild and/or untrained animals, breed-

ing soundness examinations, collection outside of the breeding season, or culling). Semen collection by EEJ has no adverse effects on ram spermatozoa's fresh or post-thaw quality. Therefore, practitioners and/or farmers should ensure the use of stress alleviative measures (anesthesia, sedatives, or hormones) while conducting the procedure. A compromise should be reached between the EEJ welfare consideration and the technique's importance in salvaging rare breeds' genetic resources. There has been remarkable progress in EEJ between the 20th and 21st centuries. The progress recorded in ram EEJ was mostly related to the following:

i.   Semen concentration (highest reported in the 21st century; $5.2 \times 10^9$/mL, compared to that recorded in the 20th century: $2.4 \times 10^9$/mL).

ii.  Animal welfare: probe type and voltage used (most of the 20th-century studies used bipolar rectal electrode probes that deliver 30 volts or probe types with ring electrodes, compared to the 21st-century studies, which mainly used three-longitudinal electrode probes, delivering a maximum of 10–15 volts).

iii. Stimulation duration (used up to 10 min in the 20th-century studies and a maximum of 5 min in the 21st-century studies).

There is a slight difference between the freezing procedures for ESR and those for AV and EEJ. Treating sheep sperm with specific mineral ions, antioxidants, or biological compounds can improve sperm motility, fertilizing ability, morula, and blastocyst rates in sheep. Lastly, poor-quality post-thawed ram semen samples can be used to achieve good fertility results with LAI or IVF.

In conclusion, the sperm collection methods discussed in the present review offer a useful tool for preserving the genetic diversity of the global sheep population. Alternative methodologies allow the users to test and adapt the sperm collection and storage methods that are the most suitable to their needs and local conditions.

**Author Contributions:** Conceptualization, S.B.; methodology, S.B., M.A.M., G.K. and S.N.; validation, not relevant; formal analysis, not relevant; investigation, not relevant; resources, M.A.M., I.E., S.B. and G.K.; data curation, not relevant; writing—original draft preparation, M.A.M.; writing—review and editing, M.A.M., S.B., G.K., S.N., I.E. and N.V.; visualization, not relevant; supervision, S.B. and N.V. All authors have read and agreed to the published version of the manuscript.

**Funding:** This research was funded by Complex rural economic development and sustainability research, development of the service network in the Carpathian Basin (EFOP-3.6.2-16-2017-00001), and the Stipendium Hungaricum Scholarship Program.

**Institutional Review Board Statement:** Not applicable.

**Data Availability Statement:** Not applicable.

**Acknowledgments:** Thanks to Somsy Xayalath for checking the manuscript for minor mistakes.

**Conflicts of Interest:** The authors declare no conflict of interest, and the funders had no role in the writing of the manuscript or in the decision in its publication.

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
