# Peer review of "Alternative Opportunities to Collect Semen and Sperm Cells for Ex Situ In Vitro Gene Conservation in Sheep"

_agriculture, doi:10.3390/agriculture12122001_

Round 1
Reviewer 1 Report
The article is a valuable study that proves a deep analysis of the problem.However, I have a few comments:
line 243 - EEJs - "s" - what does this letter mean
for example line 414 and 418 "m", 442 "s" - the full name of the word should be given
line 424-426 - sentence not fully understood in the context of the previous information.
It should be more clearly presented how the presence of cytoplasmic drops affects the quality of sperm, especially their ability to freeze.
line 432 - the author's name and the number must be provided.
I propose to combine tables 3, 4 and 5 because the results for the ESR are repeated
Author Response
Manuscript: agriculture-1996325; Mujitaba et al.
Response to Reviewer 1.
We would like to express our thanks to Reviewer 1. for taking time to review our manuscript thoroughly. Below we indicate our responses tot he comments of Reviewer 1. Comments are in italics.
Line 243 - EEJs - "s" - what does this letter mean
The sentence is rewritten - collected by EEJ
For example lines 414 and 418 "m", 442 "s" - the full name of the word should be given
done
Line 424-426 - sentence not fully understood in the context of the previous information.
Sentence is rewritten
It should be more clearly presented how the presence of cytoplasmic drops affects the quality of sperm, especially their ability to freeze.
We would direct the readers’ focus on the cited reference (88) for a deeper explanation of this phenomenon.
Line 432 - the author's name and the number must be provided.
Sentence changed.
I propose to combine tables 3, 4 and 5 because the results for the ESR are repeated
We followed the suggestion and merged Tables 3 and 4, and left Table 5 out as repeated data were shown in these as Reviewer 1. as noticed.
Besides the suggested changes, the whole text was checked by a native English speaker.
Again, we would like to thank Reviewer 1 for the comments and suggestions which clearly improved the quality and clarity of our manuscript.
On behalf of the co-authors:
Dr. István Egerszegi

Reviewer 2 Report
In this review, the authors attempted to explore the alternative methods of sheep semen collection for assisted reproductive technologies (ART) and gene conservation (GC) purposes, and provide strategies to compensate for weak or poor quality thawed sheep semen samples, so as to obtain good fertility effects. In addition, take the pre-freezing and post-thawing parameters of sperm collected by AV as a reference. Moreover; compare them with sperm collected by other semen collection methods (EEJ and ESR).
The review is interesting and valuable information about reproductive technologies. Overall, this review is a well-written and well-executed review article. Thus, I recommend acceptance with minor changes. The following are some of the minor issues.
Point1: In lines 41, 42, 43, 46, 47, 49, 53. No references are cited.
Point2: In lines 122 and 123 change the numbers to the words
Pint3: In Line 109 (this review intends to explore the alternative means of semen collection. Replace the word (means) with (methods) please.
Point4: The main body text of law review articles should be in ordinary typeface, except for case names and the titles of publications, speeches, or articles, all of which are italicized. Please double-check all the italics typos across the manuscripts.
Point5: In conclusion please summarize the main findings of this review.
Author Response
Manuscript: agriculture-1996325; Mujitaba et al.
Response to Reviewer 2.
We would like to thank Reviewer 2. for the review of our manuscript. Below we indicate our responses to the points raised by Reviewer 2. Comments are in italics.
Point1: In lines 41, 42, 43, 46, 47, 49, 53. No references are cited.
References are added.
Point2: In lines 122 and 123 change the numbers to the words
Done
Point3: In Line 109 (this review intends to explore the alternative means of semen collection. Replace the word (means) with (methods) please.
Point4: The main body text of law review articles should be in ordinary typeface, except for case names and the titles of publications, speeches, or articles, all of which are italicized. Please double-check all the italics typos across the manuscripts.
Italics typos are corrected.
Point5: In conclusion please summarize the main findings of this review.
A final conclusion sentence is added to the end of the manuscript.
Besides the suggested changes, the whole text was checked by a native English speaker.
Again, we express our thanks to Reviewer 2. for the comments and suggestions which improved the quality and clarity of our manuscript.
On behalf of the co-authors:
Dr. István Egerszegi

Reviewer 3 Report
Referee’s Evaluation Report
MANUSCRIPT IDENTIFICATION: Agriculture - 1996325
Alternative opportunities to collect semen and sperm cells for Ex-situ
In vitro gene conservation in sheep
(REVIEW)
Comments to Authors/Editor:
The paper of Mujitaba & colleagues is an interesting Review that aims to highpoint alternative ways to the artificial vagina for sperm collection; includes information regarding semen quality and some freezing techniques aligned with animal welfare claims. This manuscript falls within the scope of AGRICULTURE-MDPI. In general, the organization of this Review seems to be well structured and sustained, yet, the English quality, grammar, and sentence structure, at some points, is certainly fragile; it must be improved. After reviewing the whole manuscript, I truly believe that the Title must be rethought by the authors, based on the main components of the Review. The abstract was written in a sound fashion, mentioning the main strategies to follow in different animal production scenarios. Regarding the Introduction section, every sentence must be supported by the proper citation; in the first paragraph (i.e., L 40-53) not a single citation is included. The authors must correct this situation along with the entire manuscript. Besides, the Introduction is very confusing (i.e., L29, run-on sentence). The authors must include information regarding the importance of sheep production; the world inventory, how many breeds, at the world level, are in “danger” (i.e., endangered breeds…), from a genetic resource standpoint?. While the objectives of the study were not stated, also no working hypothesis of the study was proposed; this is a must. Despite this is not an experimental work, the authors may assume a hypothesis with respect to the main expectations of the Review based on the main findings in the scientific literature. Please include the contribution of sheep production to the world livestock sector; why do the authors decide to develop this Review based on sheep as an animal model instead of bovine, buffaloes, or goats??? Are the authors comfortable with this introduction, with no stated objectives and no working hypothesis??? L61, …,” extensive care system”????, or extensive production systems??? L62, “small/cheap meat-milk producers”???; is “cheap” an accurate word in the context of this phrase??? Is it polite to use the term “cheap producers”??? Besides, are the producers “caring animals” or “producing animals”??? Up to L71, the “Introduction” section seems to be very extended. I urgently suggest to the authors include subtitles (i.e., form L71), to better assist the readers of “Agriculture” to follow the main ideas proposed by the authors in this Review. Therefore, is fundamental to explain the “main objectives” of this Review and how the authors are going to achieve such objectives; the use of subtitles is therefore fundamental. L87, …”nail cutting”???, or disheveled, or to cut the hooves, or to remove de hooves??? It seems that the “Introduction” section goes up to L114, where “kind of objectives is described”; if so, the Introduction is really extended, and some information can be integrated into a posterior “Background section”, or “Basis or Initial methods for semen extraction”… Section 2, “Alternative semen collection methods”, and its subtitles are well developed with central information and proper citations; good work. From L198 to 202, the authors can extract or rephrase a “working hypothesis” for this Review. All the titles of the Tables must be rewritten; titles must stand by themselves in any Table or Figure. L284, g refers to grams, while g (i.e., italics), refers to the forces used in the centrifugation process (i.e., gravities???). L318; do the mentioned pregnancy rate differences reach significance???? As in section 2, sections 3 and 4, and their subtitles are well developed with key information and suitable citations. In general, the cited works and sequence of methodologies presented by the authors are relevant and in accordance with the general idea of this Review. The novelty value of this Review is reasonable. In general, the authors made an accurate interpretation of the main findings obtained from the global scientific literature. With respect to the Conclusion section, the authors must highlight the main findings obtained from the literature search in this Review. Please remember that Conclusions are not a Mini-Abstract. From my point of view, the Conclusion MUST be shortened in a significant way; this section is extremely long. Besides, conclusions must be aligned with the non-existing working hypothesis; therefore, the working hypothesis MUST be included in the Introduction section. The list of references cited in the manuscript is proper, while actualized. This is an interesting Review, with a large set of variables and methodologies discussed. Yet, on one side the authors must improve both the English language quality as well as the clarity and logical arrangement of the whole Review and the title of the manuscript must be aligned with such arrangement. It is central to ensure that the paper is readable; the authors must increase the readability and the scientific writing and merit of the manuscript, and these last ideas must be perfectly aligned with an appropriate title. All the commented issues and requests must be clearly addressed by the authors; at this point, and based on the above comments, my pronouncement is that this manuscript cannot be accepted in its actual format. It requires moderate editions and corrections.
Author Response
Manuscript: agriculture-1996325; Mujitaba et al.
Response to Reviewer 3.
The authors would like to express our sincere thanks to Reviewer 3. for the detailed review of our manuscript. Below we indicate our responses to the concerns raised by Reviewer 3. Comments are in italics.
the English quality, grammar, and sentence structure, at some points, are certainly fragile; it must be improved
The whole manuscript text was checked and proofread by a native English speaker.
I truly believe that the Title must be rethought by the authors
We still believe the title represents the work and the other two reviewers did not raise concerns, therefore we respectfully ask Reviewer 3 to accept our decision to keep the original title.
Regarding the Introduction section, every sentence must be supported by the proper citation; in the first paragraph (i.e., L 40-53) not a single citation is included. The authors must correct this situation along with the entire manuscript. Besides, the Introduction is very confusing (i.e., L29, run-on sentence). The authors must include information regarding the importance of sheep production; the world inventory, how many breeds, at the world level, are in “danger” (i.e., endangered breeds…), from a genetic resource standpoint?. While the objectives of the study were not stated, also no working hypothesis of the study was proposed; this is a must. Despite this is not an experimental work, the authors may assume a hypothesis with respect to the main expectations of the Review based on the main findings in the scientific literature. Please include the contribution of sheep production to the world livestock sector
The whole Introduction section is corrected and rewritten according tot he suggested changes – we did our best to clarify the global importance of the sheep industry; we added citations; we clarified the main reason for writing this review. Moreover, Reviewer 3. raised concerns about the improper words we chose – these were checked and corrected by a native English speaker.
All the titles of the Tables must be rewritten; titles must stand by themselves in any Table or Figure.
We rewrote the titles.
With respect to the Conclusion section, the authors must highlight the main findings obtained from the literature search in this Review. Please remember that Conclusions are not a Mini-Abstract. From my point of view, the Conclusion MUST be shortened in a significant way; this section is extremely long. Besides, conclusions must be aligned with the non-existing working hypothesis; therefore, the working hypothesis MUST be included in the Introduction section.
We shortened the conclusions and added a closing sentence that links the whole manuscript tot he newly added aim in the Introduction to present a final conclusion.
We hope the revision is sound now both from linguistic and scientific points of view.
Again, we would like to thank Reviewer 3. for the very thorough review which clearly improved the quality and clarity of our manuscript.
On behalf of the co-authors:
Dr. István Egerszegi
